# Multi-Integration Time Adaptive Selection Method for Superframe High-Dynamic-Range Infrared Imaging Based on Grayscale Information

**DOI:** 10.3390/s22114258

**Published:** 2022-06-02

**Authors:** Xingyu Tao, Weiqi Jin, Jianguo Yang, Shuo Li, Binghua Su, Minghe Wang

**Affiliations:** MOE Key Laboratory of Optoelectronic Imaging Technology and System, Beijing Institute of Technology, Beijing 100081, China; 3220190424@bit.edu.cn (X.T.); 3120195336@bit.edu.cn (J.Y.); lishuo93@163.com (S.L.); bhsu@263.net (B.S.); 3220215088@bit.edu.cn (M.W.)

**Keywords:** infrared thermal imaging, superframe multiple integration time, high dynamic range, adaptive, grayscale information, region-growing point

## Abstract

With the development of superframe high-dynamic-range infrared imaging technology that extends the dynamic range of thermal imaging systems, a key issue that has arisen is how to choose different integration times to obtain an HDR fusion image that contains more information. This paper proposes a multi-integration time adaptive method, in order to address the lack of objective evaluation methods for the selection of superframe infrared images, consisting of the following steps: image evaluation indicators are used to obtain the best global exposure image (the optimal integration time); images are segmented by region-growing point to obtain the ambient/high-temperature regions, selecting the local optimum images with grayscale closest to the medium grayscale of the IR imaging system for the two respective regions (lowest and highest integration time); finally, the three images above are fused and enhanced to achieve HDR infrared imaging. By comparing this method with some existing integration time selection methods and applying the proposed method to some typical fusion methods, via subjective and objective evaluation, the proposed method is shown to have obvious advantages over existing algorithms, and it can optimally select the images from different integration time series images to form the best combination that contains full image information, expanding the dynamic range of the IR imaging system.

## 1. Introduction

In practical infrared applications, when strong radiation targets (such as the sun, flames, or jamming bombs) exist simultaneously with ambient targets, the radiation brightness spans a large range. Furthermore, for the current 14-bit analog-to-digital (A/D) cooled IR focal plane array (IRFPA) thermal imaging systems, even without considering the non-linearity of the S-shaped response curve, the equivalent dynamic range is only approximately 84 dB [1]. The dynamic range is much smaller than that of the natural scene’s differences in radiation. This means underexposure or overexposure cannot be avoided, even if the imaging parameters of the system are adjusted. Moreover, all details of the scene cannot be captured at once, and this will have a very negative influence on several target detection and identification tasks. Hence, it is necessary to propose a high-dynamic-range (HDR) imaging method to accommodate effective imaging of the full radiation scene.

Current dynamic range extension techniques used in IR imaging can be broadly classified into pixel-based A/D conversion techniques and superframe-based variable-integration-time imaging techniques [2]. The former performs on-chip pixel-level A/D conversion on each pixel separately to achieve high-bit-width, low-noise HDR imaging, and is an advanced IRFPA design method. However, the detector process is complex and difficult to develop. In addition, the cost is high at this stage. Owing to its low-noise equivalent temperature difference (NETD), it is still challenging for the dynamic range of scenes obtained even with 18-bit A/D conversion to meet the requirements of actual HDR scenes. The latter, based on the existing IRFPA, uses variable-integration-time ultra-high-frame-rate imaging with periodic cycling to broaden the HDR IR imaging by fusing low-dynamic-range (LDR) images with different integration times [3]. It is an effective method to extend the dynamic range via digital image processing methods only.

The superframe HDR imaging technique mainly involves two parts: front-end integration time or response function adjustment; and back-end HDR fusion enhancement algorithms for LDR images. In the field of visible HDR imaging, the front-end processing mainly focuses on adaptive exposure methods based on gray scale analysis [4,5], such as gray scale histogram (including fixed chunking theory [6,7], fuzzy logic calculation weight theory [8,9], and scene area segmentation theory [10]). Evaluation methods such as edge information [11] and gradient difference [12] also play a positive role in integration time adjustment. HDR fusion methods often use certain weights to calculate the expected output of the image sequence to achieve the fusion effect [13,14,15,16,17,18,19]. For example, Mertens et al. [20] proposed a pyramidal decomposition method to evaluate LDR images using contrast, saturation, and good exposure to obtain a weight map corresponding to the three indicators for HDR fusion. Zheng et al. [21] proposed an adaptive structural decomposition method to generate a series of images using gamma correction, after segmenting the images, their contrast, expected signal output, and exposure quality under Gaussian function evaluation are calculated, and HDR fusion is finally performed. Unlike visible images, infrared images usually have low contrast, high noise, unclear details, and a more concentrated distribution of gray levels [2]. In the field of HDR infrared imaging, the integration time variation of LDR images is generally used to achieve HDR image fusion. For the front end, there is still no consensus on the selection of LDR images. Researchers often judge the response variation and exposure level roughly based on prior knowledge and the acquired LDR images, or they conduct experimental analysis of the relationship between different temperature target grayscales and medium grayscale to determine the “best” integration time [2]. However, misjudgments caused by human selection and image non-uniformity due to noise and blind spots can affect LDR image selection. Due to real-time limitations, it is difficult for imaging devices to adjust parameters quickly to a suitable exposure level. Researchers expect to adjust the integration time parameter for the next frame based on the current frame image information [22], or to adjust the relationship between exposure value and integration time [23,24] using the current image’s full pixel grayscale average compared to a preset optimal average grayscale [25,26]. Such methods have a simple processing flow and can optimize image quality to a certain extent, but they ignore local grayscales differences, and the overall results are not good. For this reason, studies on the subregions of images [27,28,29] and luminance compensation [30] have been carried out. In terms of HDR IR image fusion methods, some visible HDR fusion methods are not necessarily adapted to superframe HDR thermal imaging. HDR fusion methods based on response functions are common [1], but are more suitable for HDR thermometry applications. In addition, Li et al. [2] proposed a multi-integration time IR HDR image fusion + detail enhancement cascade algorithm for target detecting and tracking in high-dynamic-range scenes, which is a non-linear fusion method.

This paper focuses on the integration time problem of non-linear fusion of multi-integration time images in superframe HDR infrared imaging. It proposes a multi-integration time adaptive method for infrared images based on the evaluation of grayscale information and the segmentation of the region-growing point to achieve the “optimal” selection of integration time for three LDR images and combine it with the multi-integration time IR HDR image fusion + detail enhancement cascade algorithm [2] to achieve the “best” dynamic range expansion for IR imaging of HDR scenes.

The remainder of this paper is organized as follows: Section 2 presents the overall framework of the proposed method and details the two main sub-algorithms involved in this paper. Section 3 verifies the effectiveness of the proposed method by comparing the fusion results of some existing selection methods with those of this paper, and by applying the images selected by the proposed method to several typical fusion methods, which verifies the generalizability of the proposed method. Section 4 discusses the results. Finally, Section 5 concludes the study.

## 2. Materials and Methods

Typically, IR focal plane detectors have an ‘S’-shaped response curve, where the normal response of the IRFPA corresponds to a certain temperature range when the integration time varies, as in Figure 1a. When multiple LDR images with different integration times are fused, the response curves are stitched together to obtain a larger range of temperature-response output curves, as in Figure 1b. Clearly, the choices of appropriate integration time and appropriate LDR image sequence fusion method are the critical factors affecting the dynamic range and imaging quality of the fused images. If the integration time interval is small, the expansion of dynamic range performance is not obvious. If the normal response part of the response curve corresponding to each integration time is stretched in grayscales according to the mapping relationship, the “S”-shaped response curve is transformed into an approximately linear function, resulting in a fused image that is equivalent to the acquisition of a larger temperature range at one integration time; the linear extension of the dynamic range of the infrared image is achieved. This method is often used in HDR thermometric imaging, as the target’s temperature in the actual scene is often discontinuous, especially in the presence of strong radiation targets (such as the sun, interference bombs, etc.). The use of linear fusion requires a higher number of LDR images. In IR imaging, there is no need to maintain the linearity of radiation. With the use of non-linear fusion based on pixel grayscales, gradient can be used [16,17,18,19,31] to reduce the tedious process of acquiring images for thermal imaging systems.

### 2.1. Local Optimum Images Selection Based on Region-Growing Point Segmentation

For a thermal imaging camera, the imaging at low integration time extracts the information of high-temperature target regions. Thus, detailed information of an image’s ambient-temperature regions can hardly be effectively obtained. The imaging at high integration time mainly extracts the information of ambient-temperature regions, while the details of high-temperature regions will be lost because of overexposure, which significantly affects the image segmentation effect. Therefore, some classical segmentation algorithms, such as the edge detection algorithm and morphological watershed algorithm, are not entirely suitable for processing multi-integration image sequences. As can be seen in Figure 2, as the integration time increases, the high-temperature region (electric heater) gradually changes from clear to overexposed. Moreover, the ambient-temperature region (clothing folds) changes from blurred to clear and becomes somewhat distorted. This means that the integration time changes will have different effects on each region of the image, and both very high and low integration times will cause distortion of the image. In addition, the radiation brightness of objects in the same scene tends to show a jump distribution (as shown in Figure 3), and the corresponding temperature range is not easy to obtain in real time; therefore, it is difficult to perform dynamic-range stitching according to the temperature-grayscale curve.

Region-growing point segmentation [32] is a traditional algorithm that can be applied in infrared imaging. The basic method is to define the seed point and decision rules and add neighboring pixels with similar properties to the seed. The aim is to form the corresponding growth region. Analyzing the scene characteristics of HDR IR images, the grayscales in ambient-temperature regions do not differ much. In general, the difference in grayscale value between pixel point A and pixel point B in the ambient-temperature region is less than the grayscale value of any A and B pixel points, and the difference between a high-temperature region and an ambient-temperature region is often more than double the pixel value of an ambient-temperature region. Figure 4 presents the grayscale histogram distribution of the LWIR HDR thermal imaging system for a target scene imaged at three integration times. The histogram of the high-dynamic-range scene has two pronounced peaks and a large peak-to-peak distance, which can be used to segment the ambient-temperature region from the high-temperature region by performing a full scan of the image pixels. Figure 5 shows the three integration time images segmented based on a simple grayscale threshold. The integration time *T* of the thermal imaging camera is related to the number of clock cycles *n*. It is calculated as follows:(1)$$T=\frac{n}{f}$$
where *f* is the clock frequency of the circuit; the current experimental thermal imaging system 1 clock period 1/*f* is about 0.133 μs; and *n* is the current integration time corresponding to the number of clock cycles.

The histogram of an integral image is calculated as follows: The grayscales of input image *I_T_* are divided into equal intervals called HD, based on the gray level; there are a total of *M* intervals, where *M* is total number of gray levels. If the imaging system has *D* bits, then *M* is 2^D^, and the number of pixels falling in each interval is calculated as *N*. Based on the medium gray level, the *M* intervals are divided into two parts. If the system bit width is 14 bits, the medium level is 8192. The two search intervals are [0, 8192] and (8192, 16383]. In the first search interval, the maximum value of the count *N* is calculated, and the gray value corresponding to the maximum value *N* is determined as the gray value *G_L_* in the ambient-temperature region of the frame. In the last search interval, the maximum value of the count *N* is calculated, and the gray value corresponding to the maximum value *N* is determined as the gray value *G_H_* in the high-temperature region of the frame, as follows:(2)$$\left\{ \begin{array}{l} G_{L}=HD(1,x),\;\max(N(1,i)),i\in (1,\frac{M}{2}]\\ G_{H}=HD(1,y),\;\max(N(1,j)),j\in (\frac{M}{2},M] \end{array}\right.$$
where *M* is the number of histogram intervals, *N* is the number of pixels in each interval, *x* and *y* are the positions corresponding to the largest *N* values in the intervals (1, *M*/2) and (*M*/2, *M*), respectively; and *HD* are one-dimensional tuples.

According to the histogram analysis, *G_H_* represents the grayscale of most pixels in the high-temperature region, and *G_L_* represents the grayscale of most pixels in the ambient-region. The seed point (*i*_0_, *j*_0_) is determined as follows: By inputting the image at a low integration time into the image processing system, and looking for the first peak according to the histogram, the grayscale of the seed point can be determined as *G_L_*. By reading the grayscale, any pixel with a grayscale of *G_L_* is selected as the location of the seed point (*i*_0_, *j*_0_). For the same scene, the position of the ambient region stays the same, so the seed points of other images are the same as above. 

The initial value μ is set as the threshold, and the eight-connected region around the seed point is used as the growth calculation region. The pixels to be judged are divided into two regions, represented by “0” and “1” according to Equation (3). The pixels marked with “0” are classified as the seed region and considered as an ambient-temperature point; otherwise, the pixels marked with “1” are regarded as a high-temperature point, that is,
(3)$$F=\left\{ \begin{array}{l} 0,\;I\left(i,j\right)\leq \mu \times I(i_{0},j_{0})\\ 1,\;else \end{array}\right.$$
where $I(i_{0},j_{0})$ is the grayscale of the seed point, and I(i,j) is the grayscale of pixels in the eight-connected region.

After one round of the region growth operation, pixels that are marked as “0” have been added to the seed point to form a seed region. The grayscales of the seed region are averaged and used as the new seed gray value. Region growth continues with the eight-connected region around the seed region until all pixels are judged.

Unlike the general region growth method, which uses a fixed value judgement rule, the threshold μ used here is a variable. Figure 6 shows the distribution of the grayscale values for three sets of integration time IR 8-bit images of the same scene. The two markers on each graph in Figure 6 are the grayscale parameter of the ambient-temperature region and the high-temperature region, respectively. Each marker includes the coordinates of the instance points and the corresponding grayscale values. In the same scene, the difference between the grayscale in the ambient-temperature region and the high-temperature region varies with the integration time *T*, which makes the threshold μ a variable parameter rather than a non-constant parameter, denoted as μ(T). The same applies to the grayscale parameters *G_L_*(*T*) and *G_H_*(*T*). The threshold μ(T) can be obtained from the relationship between the two gray values as follows:(4)$$\mu \left(T\right)=\left[1+\frac{G_{H}(T)}{G_{H}(T)+G_{L}(T)}\right]$$

As mentioned earlier, the grayscale difference of pixels in the ambient-temperature region is not large, so the second coefficient, $\frac{G_{H}(T)}{G_{H}(T)+G_{L}(T)}$ (less than 1), is used, which determines the range from the “seed” grayscale to the high-temperature grayscale. This means that the pixels with a slightly larger grayscale than the seed point should also be considered as part of the ambient-temperature region, but this range does not exceed the grayscale of the seed point itself. When the integration time increases, the grayscale of the ambient-temperature region increases, but the high-temperature region is basically in a saturated state, with little change in grayscale, and the second coefficient decreases accordingly. It means that the range is being defined as the ambient region decreases, which avoids the error of pixels in the high-temperature region being divided into the ambient-temperature region under high integration time. As shown in Figure 7, for an image with 1995 μs, the pixels with grayscale smaller than 1.74 times that of the seed point can be defined as the ambient-temperature region, and a good segmentation effect is achieved, as shown in Figure 7b. However, if 1.2 times is used, the ambient-temperature region (e.g., the clothing) is also defined as the high-temperature region, as shown in Figure 7a; if 3 times is used, almost all pixels are defined as the ambient-temperature region, as shown in Figure 7c.

The images were divided by markers *F* into two major ranges, *R_L_*(*T*) and *R_H_*(*T*). The segmented images obtained in Figure 8 correspond to the integration time of the scene imaging sequence in Figure 2. It was experimentally verified that the image quality of the region (including sharpness, contrast, and good exposure) is centered on the average grayscale as a parabolic function, with an approximate opening downward. Part of the detail is not visible when the integration time is too high or too low, and the scene information characterized by the signal output is the most reliable at medium grayscale (see Debevec’s [33] elaboration for details). Therefore, the average grayscale of the region after segmentation according to the segmentation threshold should be as close to medium grayscale as possible. The construction of the objective function *I_be_* for image sequence selection is as follows:(5)$$I_{be}=\arg\underset{I_{T}}{\min}\left|\bar{I_{T}\left(i,j\right)}-\frac{1}{2}\times \left(2^{w}-1\right)\right|,\;(i,j)\in region$$
where *I_T_* is the image output corresponding to the integration time *T*; $\bar{I_{T}\left(i,j\right)}$ is the average grayscale of the calculated region; w is the bit width of the original data; and *region* is represented by *R_L_*(*T*) and *R_H_*(*T*), respectively.

By traversing the multi-integration sequence of images, we calculate the average grayscale of the high-temperature region of every image, $\bar{I_{T}\left(i,j\right)}$, and Equation (5) is applied to calculate the distance between average grayscale $\bar{I_{T}\left(i,j\right)}$ and medium grayscale, which means that every image corresponding to an integration time would get its corresponding grayscale distance. Then, bubble sorting is used to rank these distances and determines the closest distance to the medium grayscale; its corresponding image is called the local optimum image of the high-temperature region *I_h_*. This ensures detailed imaging of the high-temperature region. The image selection process for the ambient-temperature region is similar to the aforementioned method; the only difference is the average grayscale of the ambient-temperature region of every image, $\bar{I_{T}\left(i,j\right)}$, is calculated, which obtains the result called the local optimum image of the ambient-temperature region *I_l_*. That is, when choosing *I_h_*, the high-temperature region *R_H_* is applied in “*region*”; when choosing *I_l_*, the ambient-temperature region *R_L_* is applied in “*region*”.

### 2.2. Best Global Exposure Image Selection Based on the Information Evaluation

#### 2.2.1. Selection of Image Quality Evaluation Indicators

The evaluation of the image quality is multidimensional. If an image is evaluated from only one dimension, it will face the problem of a non-universally applicable evaluation method. Therefore, it is more reasonable to use multiple indicators to evaluate the image comprehensively.
(1)Information entropy

In the field of informatics, all pixels in an image can be regarded as discrete sources of information. Therefore, the information entropy is defined as follows: Suppose the image is an *m* × *n* matrix of pixels, and each pixel corresponds to a gray level between 0 and ($2^{w}$ − 1). Moreover, suppose the probability of each gray level to be *P_i_*.
(6)$$P_{i}=\frac{\alpha_{i}}{m\times n}$$
where *α_i_* is the sum of the pixels of each gray level. The information entropy is expressed as follows:(7)$$H(X)=\sum_{i=0}^{{}^{2^{w}-1}}P(X_{i})\log\left[P(X_{i})\right])$$
where w is the system bit width, and *X_i_* is the *i*th gray level.

According to the analysis, if the exposure time is very long or short, the quality of the image obtained will be reduced. The information capacity of the image and the value of the information entropy are both reduced. As shown in Figure 9, the relationship between the information entropy and integration time is a parabolic function with a unique extremum point. In addition, the extremum point is monotonic at both ends. If the exposure time is appropriate, the image will have the highest value of information capacity, as well as information entropy.
(2)Image gradient

In mathematics, gradient is a concept of change rate: the gradient of an image characterizes the degree of the abrupt change in the target edge, effectively measuring the amount of edge detail. The gradient of each point of an image is a two-dimensional vector comprising partial derivatives in the *x* and *y* directions. From its definition, the gradient is only applicable to calculating continuous functions. Thus, it is appropriate to use finite difference for a discrete function of two dimensions, such as an image. The original image is convolved with the *S* operators *h_x_* and *h_y_* to obtain the images *I_x_* and *I_y_*, respectively, and the absolute values of *I_x_* and *I_y_* are calculated to obtain the gradient sum.
(8)$$S=\sum_{M}\sum_{N}\left(\left|I_{x}\right|+\left|I_{y}\right|\right)$$
where the *S* operators *h_x_* and *h_y_* are as follows:(9)$$h_{x}=\left[\begin{array}{ccc} -1 & 0 & 1\\ -2 & 0 & 2\\ -1 & 0 & 1 \end{array}\right],\;h_{y}=\left[\begin{array}{ccc} -1 & -2 & -1\\ 0 & 0 & 0\\ 1 & 2 & 1 \end{array}\right]$$
(3)Average grayscale and grayscale average variance

Average grayscale *I_mean_* is a common image quality evaluation indicator. Considering the average of the gray levels of all the pixels in an image will give an overall analysis of what level of gray the image is at. It is calculated as follows:(10)$$I_{mean}=\frac{1}{NM}\sum_{i=1}^{N}\sum_{j=1}^{M}I(i,j)$$
where *N* and *M* are the width and height of the image, respectively.

If the average grayscale of the image is much smaller than that of the medium grayscale, the grayscale histogram is to the left, indicating that the integration time is too small for the high-temperature region to be imaged effectively; if the average grayscale is close to the saturation grayscale, the grayscale histogram is to the right, indicating that the integration time is very high, and the ambient-temperature region is close to overexposure.

The grayscale average variance *I_std_* is the sum of the differences between the grayscale values of all of the image’s pixels and the average grayscale value of the image, calculated as follows:(11)$$I_{std}=\frac{1}{MN}\times \sum_{i=1}^{N}{\sum_{j=1}^{M}\left[I\left(i,j\right)-I_{\mathrm{mean}}\right]}^{2}$$

If the grayscale average variance is large, the gray values of each pixel of the image are widely distributed. The grayscale histogram has a large and flat span of gray levels, and the image obtained has a large dynamic range and more details.

#### 2.2.2. Selecting the Best Global Exposure Image Based on Image Quality Evaluation Indicators

Typically, images under natural imaging cannot achieve the best of all indicators; the aforementioned four image evaluation indicators at their best may correspond to different images of a multi-integral image sequence. Therefore, a ranking method needs to be determined to select the best global exposure image.

The information entropy indicator is sorted from the largest to the smallest. The image gradient indicator is sorted by absolute value, from largest to smallest. The average grayscale indicator is sorted from the smallest to the largest difference from the medium grayscale. The grayscale average variance indicator is sorted from the largest to the smallest. Finally, the above rankings are summed to obtain the comprehensive ranking of all image evaluation indicators. The image with the first comprehensive ranking is obtained, which is considered to be the best global exposure image, as shown in Equation (12).
(12)$$\{\begin{array}{c} Mul_{rank}=SUM(H_{rank},\;S_{rank},\;M_{rank},\;Std_{rank})\\ \max(Mul_{rank})\to Integral_{opt} \end{array}$$
where *Mul_rank_* is the comprehensive ranking of the image calculated as the sum of the rankings of each evaluation indicator; *H_rank_* is the ranking of the information entropy *H*; *S_rank_* is the ranking of the image gradient *S*; *M_rank_* is the ranking of the image average grayscale *I_mean_*; *Std_rank_* is the ranking of the image grayscale average variance *I_std_*; and *Integral_opt_* is integration time corresponding to the maximum of *Mul_rank_*.

Taking the images involved in Figure 2 as an example, the ranking of each image evaluation indicator is shown in Table 1. An image with a clock period of 15,000 can be used as the optimal global image for the scene. The best results of the evaluation indicators have been marked in bold.

Owing to some problems, such as thermal camera processing, the background of the original image is noisy, and the problem of bad pixels is prominent. Thus, the image needs to be preprocessed before fusion, including the removal of blind elements, using a blind element interpolation algorithm, noise filtering, and two-point correction for nonuniformity correction.

### 2.3. Image Fusion and Enhancement

In this study, the multi-integration time IR HDR image fusion + detail enhancement cascade algorithm proposed by the research team [2], called MIF&DE, is used to fuse and enhance the images. The steps are as follows:

**Step 1.** Input image sequences, calculate mean grayscale of each image, and find the image closest to the medium grayscale, which will be regarded as the best-exposed image *I_b_*_1_.

**Step 2.** Locate the strongly radiated region in the best-exposed image using threshold segmentation.

**Step 3.** Calculate the mean gray level value of the strongly radiated region in the image sequence, find the image closest to the medium gray level, and consider it as the image *I_b_*_2_ with the best exposure to the strongly radiated region.

**Step 4.** Calculate the gradient maps of images *I_b_*_1_ and *I_b_*_2_ and map them to their respective target gradient map, finally form a synthetic target gradient map.

**Step 5.** Obtain the weight values of each image sequence according to the Gaussian function and form the target gray map according to the weights.

**Step 6.** Finally, the objective function is constructed based on the target gradient map and the target gray map, and minimize the objective function to obtain the fused image.

**Step 7.** Input the fused image, divide the image blocks into simple as well as complex blocks by machine learning, and merge the simple blocks into simple regions.

**Step 8.** Calculate the grayscale mapping table of the simple and complex blocks according to the critical visible deviation of the human eye. Finally, calculate the output grayscale to obtain the enhancement effect of the fusion image.

For HDR imaging, the high-temperature/low-temperature target scene region and the ambient-temperature region need to be studied separately; except for the local effect, the continuity of the entire scene should also be considered. Therefore, we propose a multi-integration time adaptive selection method based on grayscale information, shown in Figure 10. The whole image processing process is divided into four modules: the image acquisition module, the integral image sequence selection module, the image fusion module, and the image enhancement module. The integral image sequence selection module involves the selection of local optimum images and the selection of the best global exposure image, and it is worth noting that the two processes can be carried out in parallel or vertically.

**Step 1.** Adjustment of different integration times to capture the infrared image sequence.

**Step 2.** Input to the integral image sequence selection module.

**Step 3.** Select the local optimum images using region-growing point segmentation. Iterate through the integration image sequence to calculate the grayscale histogram and determine the segmentation threshold; perform region segmentation in high-temperature region *R_H_*(*T*) and ambient-temperature region *R_L_*(*T*); and find the respective optimal objective function to determine the local optimum image of ambient-temperature region *I_l_* and local optimum image of high-temperature region *I_h_*.

**Step 4.** Select an image with the best overall evaluation index as the best global exposure image using the grayscale information evaluation index, which can compensate for the absence of certain details in the local best exposure image.

**Step 5.** Input the three selected images *I_l_*, *I_h_*, and *I_m_* into the image fusion module.

**Step 6.** Input the fused HDR image into the image enhancement module to obtain a visualized output.

## 3. Experimental Comparison Results

To verify the effectiveness and generalizability of the algorithm, two thermal imaging cameras with different performance were chosen to capture different scenes. InfraTec’s cooled mid-wave thermal imaging camera ImageIR8355 (shown in Figure 11a), made in Germany, was used as imaging instrument 1, with the following specifications: MCT detector type, spectral range 3.7–4.8 μm, 640 × 512 pixels, NETD ≤ 20 mK, and 14-bit dynamic range. Cedip’s cooled mid-wave thermal imaging camera Jade (shown in Figure 11b), made in France, was used as imaging instrument 2, with the following specifications: detector type InSb, spectral range 3–5.2 μm, 320 × 256 pixels, NETD ≤ 25 mK, and 14-bit dynamic range.

A blackbody-based nonuniform correction is required prior to the operation of the thermal imaging camera to eliminate the differences in the response of the detector image elements. The specific operation process involves using the thermal imaging camera to visualize the blackbody. To ensure the blackbody is in a stable working condition, its temperature should not be much less than the ambient temperature, nor should it exceed 100 °C. The blackbody temperature should be controlled, starting from 10 °C and gradually increasing to 70 °C, in 5 °C steps. The upper and lower thresholds of the integration time depend on the thermal imaging camera. When most of the pixels of the blackbody image are saturated, the integration time has reached the upper limit; when the blackbody image cannot be displayed, the integration time has reached the lower limit. The blackbody sequence images at different temperatures and integration times were acquired. The mean grayscale value of a single image was calculated. The temperature-grayscale curves corresponding to different integration times were fitted using MATLAB software, where temperature and grayscale were approximately proportional, and a straight-line fitting method was chosen. Next, we found the two temperature values that fit best and considered their corresponding mean grayscale values as the standard grayscale values, so that the correction coefficients for each pixel could be calculated. The gain and bias coefficients are required for the pixel to obtain the corresponding standard grayscale values. In addition, outliers resulting from pixel saturation need to be removed. As shown in Figure 12, if the saturated image produced by excessively high temperature is not eliminated, the linear fitting in Figure 12a will select the two temperatures with the highest fitting degree, 15 °C and 35 °C, denoted as (15, 35); after removing the saturated image, the linear fitting at this integration time becomes that seen in Figure 12b, and the two temperatures with the highest fitting degree, 15 °C and 40 °C, are selected, denoted as (15, 40). The cooled mid-wave thermal imaging camera ImageIR8355′s integration time sequence is: 5, 8, 11, 13, 20, 27, 40, 53, 67, 80, 93, 106, 133, 200, 266, 332, 399, 466, 532, 665, 798, 931, 1064, 1197, 1330, 1596, 1862, 2128, and 3990 μs. The cooled mid-wave thermal imaging camera Jade’s integration time sequence is as follows: 50, 100, 200, 300, 400, 500, 600, 700, 800, 900, 1000, 1100, 1200, 1300, and 1400 μs.

Three thermal images are selected using the proposed method and processed by denoising and blind element rejection, and the correction coefficients are selected according to the corresponding integration time or the closest integration time for correction processing [1]. Fusion and enhancement are performed using multi-integration time IR HDR image fusion + detail enhancement cascade algorithm. To demonstrate the effectiveness of the proposed method, four existing methods were selected for comparison.
(1)The empirical selection method [2], called ES, selects four different temperatures of objects to be imaged separately and adjusts the integration time so that the average grayscale value is closest to the medium grayscale value to determine the four integration times.(2)The image-evaluation-indicator-based method, called EI, selects image-based image evaluation indicators only. The selection principle is that the information entropy, gradient difference, average grayscale, and average grayscale square difference of each image in the sequence are calculated, and the best image corresponding to each of the four indicators is selected for fusion. The calculation results may have overlapping parts (as shown in Table 2).(3)The maximum gradient difference method [12], called MGD, calculates the Laplace gradient for the image sequence, and the two images with the greatest gradient difference are selected as the fused images.(4)The multi-region mean weighting + information entropy selection method [29], called MW&EN, adjusts the integration time of the next image frame by calculating the grayscale weighting of different regions so that the average grayscale is close to the desired mean value, and when the adjustment appears to be overshot, the integration time is adjusted using the maximum information entropy to finally obtain the image corresponding to the best integration time.(5)The maximum information entropy adjustment method [27], called ME, uses a histogram to segment the image, and searches for the integration time corresponding to the maximum information entropy in each region.

Table 2 gives the selected integration time of the five methods. After thermal imaging, selection, fusion, and enhancement processing, the final results of the three sets of scenes are shown in Figure 13, Figure 14 and Figure 15.

To further illustrate the generalizability of the proposed method, the images selected by the proposed method need to be fused with different fusion methods. The proposed method is applied to several typical fusion methods, including those proposed by Mertens [20], Vonikakis [34], Ma [35], and Kou [17], as well as MIF&DE [2], as shown in Figure 16, Figure 17 and Figure 18.

## 4. Discussion

From a subjective evaluation point of view, the single-integration images are not able to visualize the high-temperature region or the ambient-temperature region well in any of the three scenes. For the ES method, scene 1 was visualized well. This is because the integration time was selected using four groups of different temperature targets for that scene (as opposed to one for the other scenes). The temperature was also changed, which could cause the selected integration time to fail. In addition, different thermal imaging cameras have different imaging performances for the same target, so the integration time will also be different. For example, the high-temperature heater in Scene 3 has been distorted by “black edges”. For the EI method, a common problem is overexposure in high-temperature regions. For the MGD method, the serious problem is that the ambient-temperature regions will be blurred. This phenomenon exists in scene 1 and scene 2: the electric heater is distorted in scene 2, and although there is a good overall imaging effect in scene 3, the halo of the door is strong, and the outline of the bicycle behind the figure needs to be deepened. For the MW&EN method, the high-temperature region has poor imaging quality in each scene. For the ME method, the details in ambient-temperature areas are visualized well; however, the high-temperature regions have poor quality, such the heater in scene 2 has some distortion. It can be seen that the fused image obtained by the proposed method takes into account the details of the high-temperature region and the ambient-temperature region; there is no image distortion phenomenon, and a better image quality is obtained.

As there is no accepted indicator for evaluating the quality of HDR IR fusion images, we used roughness (*ρ*), fused visual information fidelity (VIFF) [36], natural image quality evaluation indicator (NIQE) [2], and information entropy (Entropy) to further examine the proposed method in terms of noise level, fidelity, visual perceptual quality, and average information, respectively. In general, the lower the value of *ρ*, the smoother the image and the lower the noise level of the image; the larger the VIFF, the more accurately the fused image retains the visual information of the source image and the better the image quality; the smaller the NIQE, the better the visual perceptual quality of the image; and the larger the Entropy, the richer the detail information contained in the image. Table 3 gives the fusion image evaluation indicators for scene 1, scene 2, and scene 3. The best results of the evaluation indicators have been marked in bold.

From an objective evaluation point of view, due to the special nature of the multi-region mean-weighted + information entropy selection method, which predicts the next image based on the current frame, the obtained image has no control over the original image, so the VIFF indicator of this method is not relevant. As the MGD method picks two images, there is a possibility of less image information, which leads to a blurred fused image and reduced roughness, which is the reason why the MGD method has the lowest *ρ* value. In addition to MGD, the proposed method has the lowest roughness and the smoothest images among the remaining methods. After testing several scenes with indicator averaging, the proposed method shows the best performance in VIFF value, NIQE value, and Entropy value, which shows that the proposed method has certain effectiveness.

As for the result of the typical fusion methods mentioned above, although there is some distortion in the fusion result of Vonikakis ‘method in scene 3 (in the electric heater), most of the methods achieved good image quality at an overall level. By comparing the imaging results, Mertens’ proposed method shows better results in both the high-temperature region and the ambient-temperature region, with details being clearly visualized. The fusion method proposed by Mertens is recommended as a follow-up fusion method to the proposed method in this paper. The high-temperature region and the ambient- temperature region can both be identified, which demonstrates the universality of the proposed method. In scene 2, a blur shows up in the man’s glasses; this is because the man moved when the integration time changed, and it does not affect the validation of the proposed method.

## 5. Conclusions

The proposed method aims at fusing IR images using a variable-integration-time ultrahigh-frame-rate imaging technique. Based on the grayscale distribution of the scene, the high-temperature region and the ambient-temperature region are marked, and the two regions with a grayscale closest to the medium grayscale are identified, respectively. The image sequences are evaluated according to the image evaluation criteria to select the best exposure image. Finally, the selected images are fused and enhanced. The fusion effect of the selected images is evaluated against other selected methods. Thus, the experimental results demonstrate the feasibility of the proposed method.

The biggest advantage of the proposed method is that it solves the problem of difficult-to-visualize high-dynamic-range infrared scenes clearly. Current infrared imaging techniques are good for ambient-temperature regions, but tend to ignore details in the high-temperature regions. Obviously, the proposed method is better for the high-temperature region, and takes into account the ambient-temperature region, selecting the images that contain the maximum information of the multi-integration sequence at the early stage of imaging to achieve a better overall imaging effect. The method integrates the image-selection stage with the subsequent image-fusion-enhancement algorithm. This proposed method can achieve the integration time image sequence input and the fusion image enhancement display output. The algorithm is better performing. However, in terms of the selection of the threshold value for the division of the high-temperature region, it still needs to be adjusted according to different scenes. This practice will be time consuming. The adaptive aspect needs to be further improved.

## Figures and Tables

**Figure 1 sensors-22-04258-f001:**
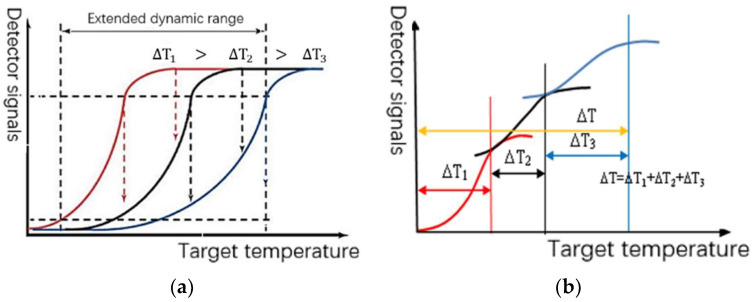
LDR response curve of IRFPA and its variable integration time expansion. (**a**) S-shaped response curve of IR imaging system; (**b**) expansion of response function with variable integration time.

**Figure 2 sensors-22-04258-f002:**
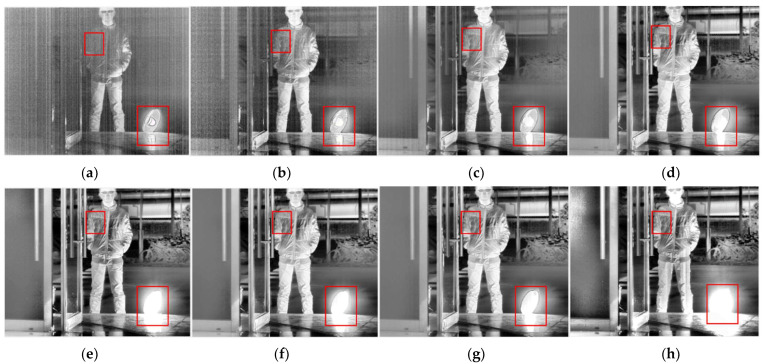
Integration time image sequence plot of high-dynamic-range scene (enhanced effect): (**a**) 13 μs; (**b**) 67 μs; (**c**) 133 μs; (**d**) 332 μs; (**e**) 665 μs; (**f**) 1330 μs; (**g**) 1995 μs; (**h**) 3990 μs.

**Figure 3 sensors-22-04258-f003:**
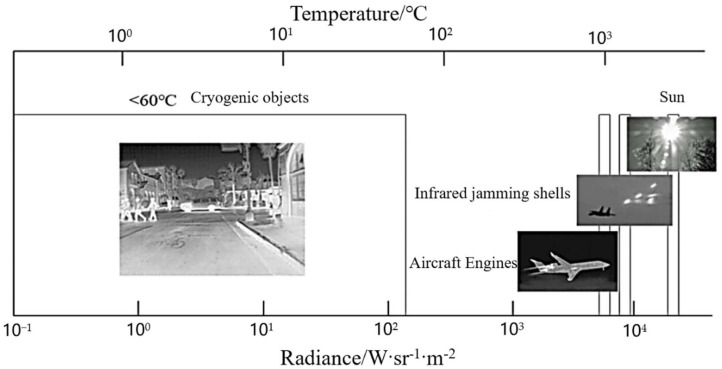
Radiance distribution of typical objects at different temperatures in the LWIR band.

**Figure 4 sensors-22-04258-f004:**
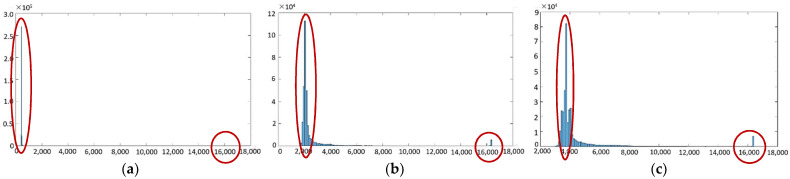
Histogram of the grayscale distribution of single-integration time-based IR images: (**a**) 13 μs; (**b**) 665 μs; (**c**) 1330 μs.

**Figure 5 sensors-22-04258-f005:**
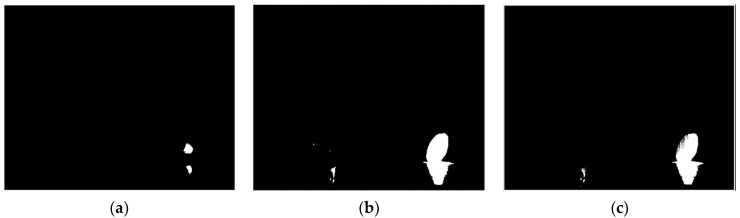
Simple segmentation based on grayscale thresholding: (**a**) 13 μs; (**b**) 665 μs; (**c**) 1330 μs.

**Figure 6 sensors-22-04258-f006:**
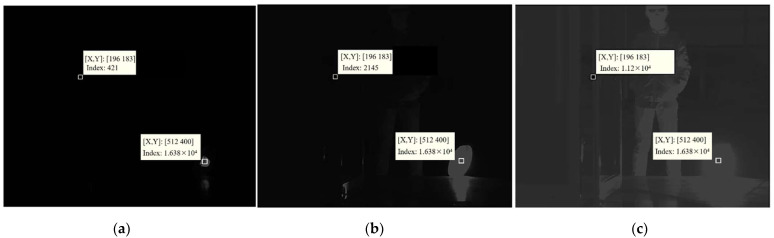
Distribution of gray levels in a set of scene maps: (**a**) 13 μs; (**b**) 665 μs; (**c**) 1330 μs.

**Figure 7 sensors-22-04258-f007:**
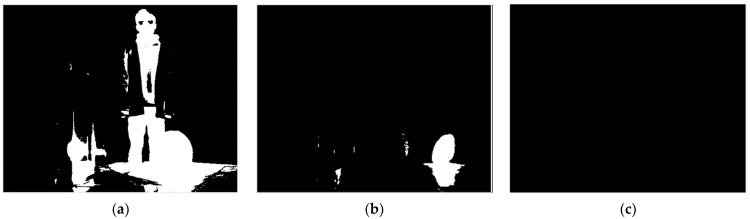
Segmentation results of different segmentation parameters at integration time 1995 μs: (**a**) μ(1995) = 1.20; (**b**) μ(1995) = 1.74; (**c**) μ(1995) = 3.00.

**Figure 8 sensors-22-04258-f008:**
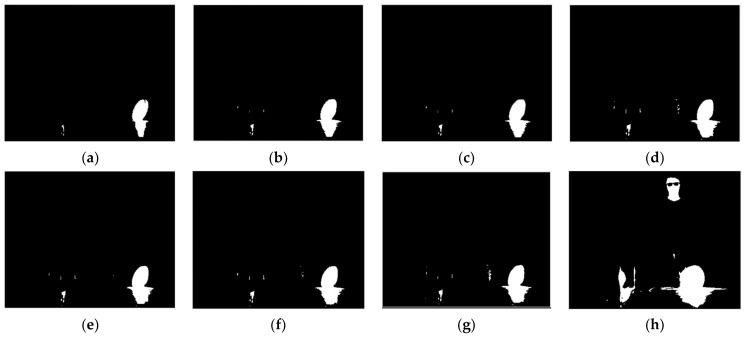
Image sequence obtained based on region-growing point segmentation: (**a**) 13 μs; (**b**) 67 μs; (**c**) 133 μs; (**d**) 332 μs; (**e**) 665 μs; (**f**) 1330 μs; (**g**) 1995 μs; (**h**) 3990 μs.

**Figure 9 sensors-22-04258-f009:**
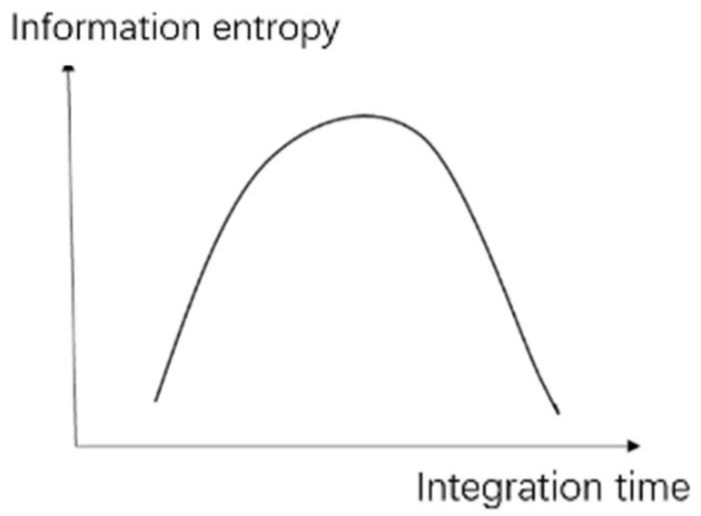
Relationship between information entropy and exposure time.

**Figure 10 sensors-22-04258-f010:**
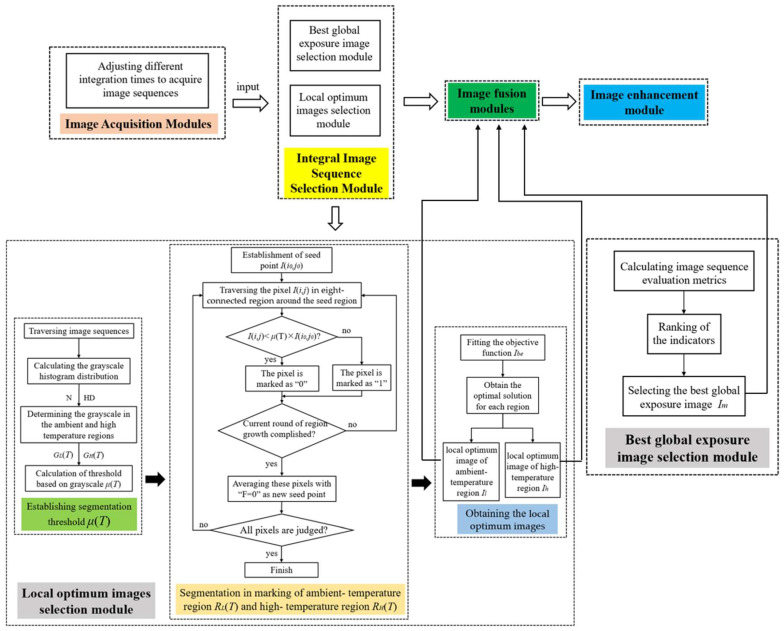
Flowchart of multi-integration time adaptive selection method for superframe high-dynamic-range infrared imaging based on grayscale information.

**Figure 11 sensors-22-04258-f011:**
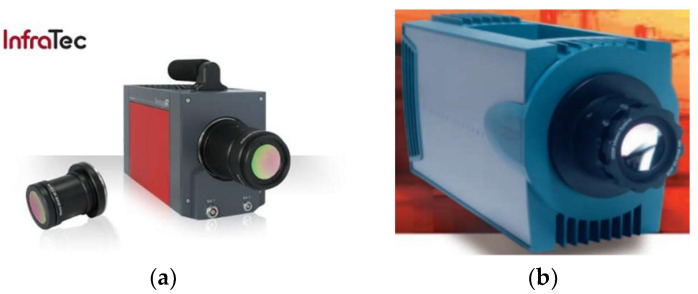
Experimental mid-wave IR thermal imaging system. (**a**) Cooled mid-wave thermal imaging camera ImageIR 8355; (**b**) cooled mid-wave thermal imaging camera Jade.

**Figure 12 sensors-22-04258-f012:**
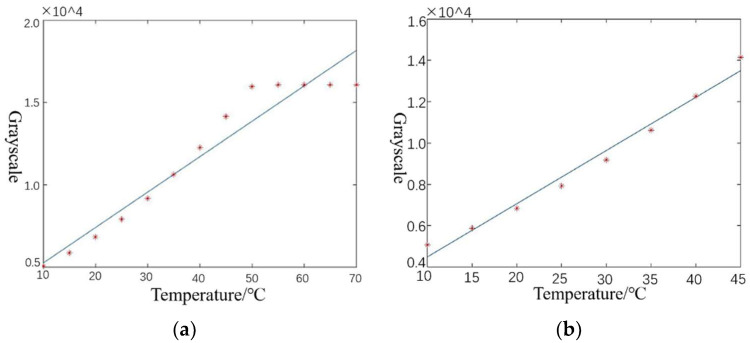
Fitting curve for two-point correction parameter: (**a**) fitted curve without removal of overexposure outliers; (**b**) fitted curve with removal of overexposure outliers.

**Figure 13 sensors-22-04258-f013:**
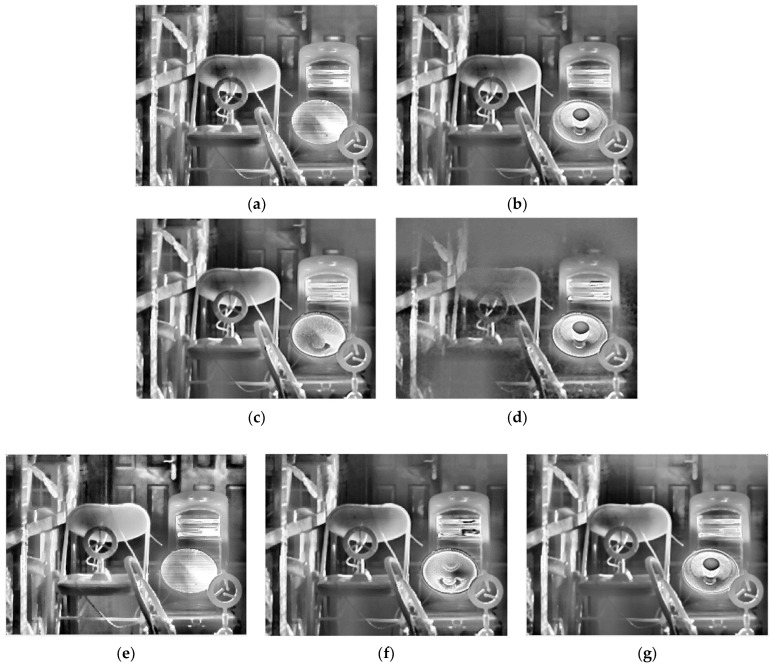
HDR fusion results for scene 1 with different selection methods: (**a**) single integration time (1050 μs); (**b**) ES; (**c**) EI; (**d**) MGD; (**e**) MW&EN; (**f**) ME; (**g**) proposed.

**Figure 14 sensors-22-04258-f014:**
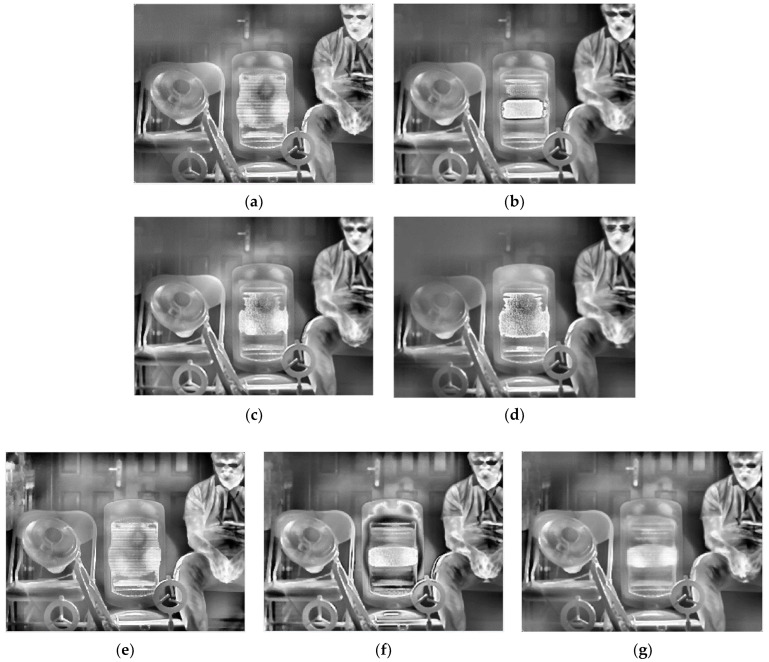
HDR fusion results for scene 2 with different selection methods: (**a**) single integration time (900 μs); (**b**) ES; (**c**) EI; (**d**) MGD; (**e**) MW&EN; (**f**) ME; (**g**) proposed.

**Figure 15 sensors-22-04258-f015:**
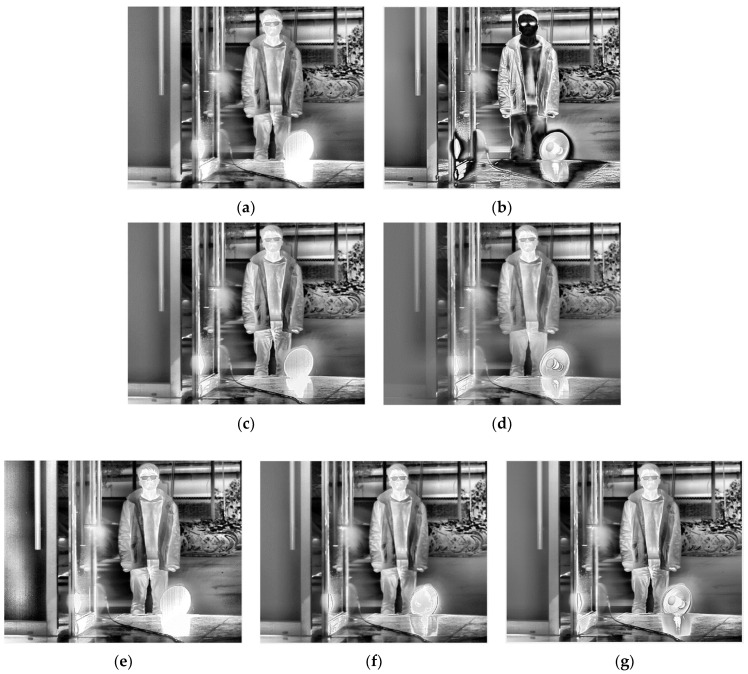
HDR fusion results for scene 3 with different selection methods: (**a**) single integration time (1995 μs); (**b**) ES; (**c**) EI; (**d**) MGD; (**e**) MW&EN; (**f**) ME; (**g**) proposed.

**Figure 16 sensors-22-04258-f016:**
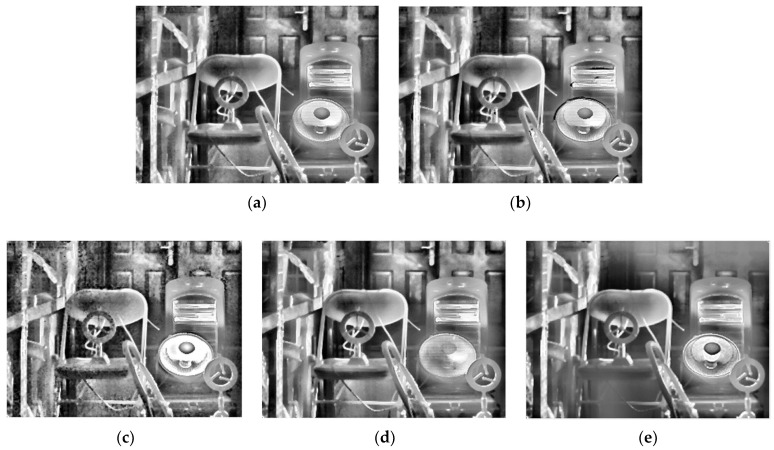
Fusion results of five typical fusion methods for scene 1: (**a**) Mertens; (**b**) Vonikakis; (**c**) Ma; (**d**) Kou; (**e**) MIF&DE.

**Figure 17 sensors-22-04258-f017:**
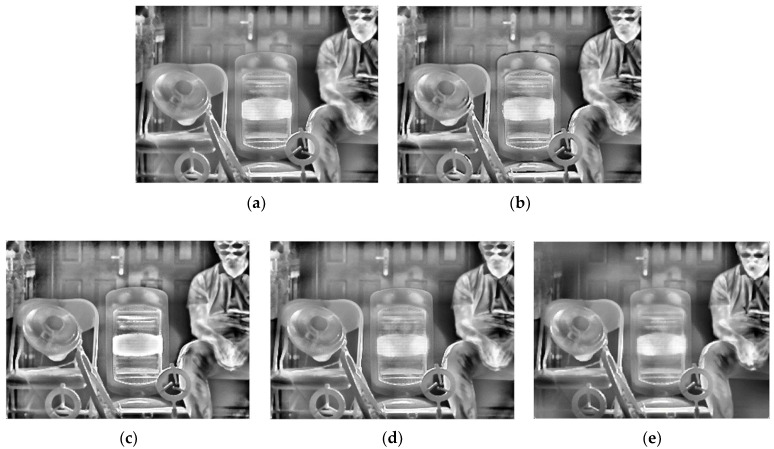
Fusion results of five typical fusion methods for scene 2: (**a**) Mertens; (**b**) Vonikakis; (**c**) Ma; (**d**) Kou; (**e**) MIF&DE.

**Figure 18 sensors-22-04258-f018:**
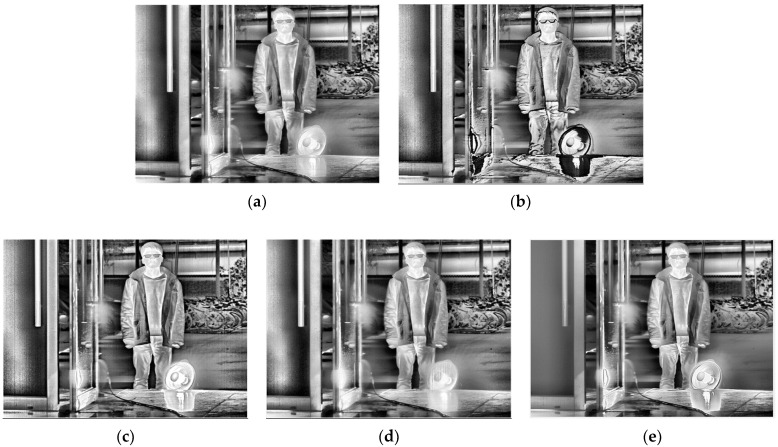
Fusion results of five typical fusion methods for scene 3: (**a**) Mertens; (**b**) Vonikakis; (**c**) Ma; (**d**) Kou; (**e**) MIF&DE.

**Table 1 sensors-22-04258-t001:** Ranking of the four image evaluation indicators for a given scene.

Integration Time/μs	*H*		*S_obel_*		*M_eanI_*		*S_tdI_*		*Mul_rank_*	
	*Value*	*H_rank_*	*Value*	*S_rank_*	*Value*	*M_rank_*	*Value*	*Std_rank_*	*Value*	*Mul_rank_*
13	3.689	➇	1516	➇	479	➇	658	➇	32	➇
67	4.933	➆	9628	➆	696	➆	1105	➆	28	➆
133	5.798	➅	21,613	➅	930	➅	1413	➅	24	➅
333	6.959	➄	56,978	➄	1567	➄	1839	➄	20	➄
665	7.834	➃	111,145	➃	2540	➃	2019	➂	15	➃
1330	8.728	➂	233,363	➂	4431	➂	2140	➁	11	➂
1995	9.221	➁	354,572	➁	**6274**	**➀**	**2262**	**➀**	**6**	**➀**
3990	**9.532**	**➀**	**831,793**	**➀**	11,496	➁	1992	➃	8	➁

**Table 2 sensors-22-04258-t002:** Integration times selected for the five methods in the three scenes.

Methods	Integration Time/μs		
	Scene 1	Scene 2	Scene 3
**ES**	50, 500, 800, 1150	50, 500, 800, 1150	67, 665,1330, 1995
**EI**	200, 350, 1050, 1400	600, 650, 900, 1400	665,1995, 3990
**MGD**	50, 350	550, 1000	13, 665
**MW&EN**	1100	900	1995
**ME**	150, 1150	250, 1200	133, 1330
**Proposed**	50, 1050, 1150	400, 900, 1200	67, 1995, 3990

**Table 3 sensors-22-04258-t003:** Evaluation results of image indicators for the three groups of scenes.

Indicators	Comparison Chart	Scene			
		1	2	3	Mean Value
** *ρ* **	ES	0.0713	0.0595	0.0499	0.0602
	EI	0.0692	0.0590	0.0538	0.0607
	MGD	**0.0550**	0.0563	**0.0434**	**0.0516**
	MW&EN	0.1674	0.1516	0.1464	0.1551
	ME	0.1110	0.1402	0.1247	0.1253
	Proposed	0.0672	**0.0537**	0.0542	0.0584
**VIFF**	ES	0.5602	0.5098	0.6171	0.5624
	EI	0.5143	0.4919	0.4632	0.4898
	MGD	0.5625	**0.6592**	0.4766	0.5661
	MW&EN	\	\	\	\
	ME	0.5858	0.4105	0.6380	0.5448
	Proposed	**0.5955**	0.6213	**0.6491**	**0.6220**
**NIQE**	ES	5.6093	5.2954	4.0974	5.0007
	EI	5.9566	5.4424	3.7536	5.0509
	MGD	6.2177	6.6463	3.786	5.5500
	MW&EN	6.2698	6.9657	**3.0805**	5.4387
	ME	5.3496	4.8063	3.7259	4.6273
	Proposed	**4.8099**	**4.7036**	3.7427	**4.4187**
**Entropy**	ES	9.3389	10.9759	10.9916	10.4355
	EI	9.2585	10.9844	10.8082	10.3504
	MGD	8.1782	10.8117	9.5601	9.5167
	MW&EN	7.8554	7.7013	7.8682	7.8083
	ME	9.4477	10.7024	10.6744	10.2748
	Proposed	**9.6904**	**11.3003**	**11.0219**	**10.6709**

## Data Availability

Not applicable.

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
