# Peer review of "Multi-Integration Time Adaptive Selection Method for Superframe High-Dynamic-Range Infrared Imaging Based on Grayscale Information"

_sensors, 2022, doi:10.3390/s22114258_

Round 1

Reviewer 1 Report

Brief expert opinion

on a manuscript submitted to sensors journal (sensors-1707698)

“Multi-integration time adaptive selection method for super-frame high dynamic infrared imaging based on grayscale information”

Authors: X. Tao, W. Jin, J. Yang, S. Li, B. Su, M. Wang

The presented manuscript describes an algorithm to generate high-dynamic range (HDR) infrared images from a series of variable integration time low-dynamic range (LDR) infrared images claiming to tackle the integration time problem.

The present manuscript is not suitable to be accepted for publication due to its deficiencies both in the way the work is presented as well as the in the derivation and justification of the derived algorithm.

The English language of the manuscript needs to be revised by a natively English-speaking human to eliminate the few but non negligible errors in the text. E.g. the meaning of the word “edges” used in conjunction with the gray level distribution is neither intuitive to the reader nor is a valid redefinition of that word given in the manuscript. Another example is the superlong sentence spanning from line 112 to line 121, which needs to be split into smaller sentences. Other such types of sentences are in the manuscript.

The labels on the graphs often are too small and illegible. The labeling in Figure 1 is inconsistent between 1a and 1b (\delta t_{int}(1) vs \Delta T_1)and to the text (t ).

The variables i and j are not explained in the text, neither they are used consistently throughout the text (sometimes they are used, sometimes not).

Some of the variables in Figure 2 are introduced only later in the text thus rendering the reader unable to understand Figure 2 at first. I_ben and I_beh are not directly referenced in the text.

There is no literature reference to the “traditional” “region-growing point segmentation” (line 174), thus the reviewer could not establish it’s correct usage nor its appropriateness.

Reconsider renaming the variables for integration time and clock frequency, which in other contexts usually are T and f rather than t and T.

Although vitally important for the algorithm, the selection of i_0 and j_0 is neither mentioned nor explained.

Although \mu is considered a variable parameter, no variable is given, neither as variable \mu(x) nor as index \mu_x. The reviewer understands, that the parameter should be the integration time or integration time index. The same applies to the given LDR images, which should have a integration time dependent parameter, which is missing throughout all equations.Thus, it remains unclear to which LDR image the equations have to be applied.

Where is no engineering or scientific reason given for the 50% cirterium at which the gray level intervals have to be split (line 223).

The variable L is either used as two-dimensional image (L_range) or as single component value (L_gray). That is inconsistent, and thus one more example why the manuscript is really hard to understand.

There is not justification given for selecting the threshold to \mu = H_gray / L_gray. Is this arbitrary, what happened if it is choosen higher or lower?

There is no description and equation for I_l and I_h.

Axis of figure 11 are unlabeled.

It is not clear how to choose the integration time for the low and high exposure time LDR images as given in Table 2, last row.

It is not clear why the proposed algorithm yields better results althrough the ambient image details are much weaker compared to MW&EN in figure 12-14.

There is no clear advise on which algorithm to choose between the Mertens, Vonikakis, Ma, Kou and MIF&DE in figure 15-17.

Roughness \rho is not lowered for the proposed methods in table 3 against your claim in line 518.

The advantages of the algorithm have to be clearer outlined.

Reviewer 2 Report

This paper proposed a novel fusion method for super-frame high dynamic infrared image. The proposed method achieves better performance according to the limited experiments. However, I still have some concerns.

1. The authors should improve the flow chart of this method for easy to understand how the proposed method work.

2. methodology? What further controls should be considered?

3. Some figure captions should rewrite and double check. such as The captions in Fig.12-14 and Fig.15-16 are not consist, the authors should give some explanations.

4. Authors should add more comparative experiments with the latest published methods.

Round 2

Reviewer 1 Report

The english language of the manuscript has improved, just introduce IR, LWIR and HD somewhere.

However, still the technical explanation of the algorithm not sufficient for a scientific publication.

Although "region-growing point segmentation" is mentioned, it is not used in the algorithm. Instead, from the given manuscript the reader has to conclude that only simple threshold segmenation was applied.

In lines 162-163 the authors assume that the edges of the images represent mostly ambient-temperature regions. Is that assumtion justified? Since also high-temperature regions may be at the image edges, is this assumption a requirement?

Equation 1 should be T = n/f.

Still the manuscript lacks information on how to just select the seed point (i0,j0) (line 189).

What is a "non-numeric parameter" (line 201)?

What is the selection parameter "scenarios x"?

Line 210: Divide an image into intervals? Or do you mean the image's grayscales?

How many gray level intervals M have to be choosen (line 213)?

Equation 3: "x" amiguously used (scenario and histogram index).

Using µ as defined by equations 2 and 3 will result in a threshold which propably splits the high-temperature grayscales. Is this intentional? The splitpoint then crucially depends on the selection of (j0,i0), if I(j0,i0)>G_L then segmention might result in large undersegmentation.

What's then meaning of figure 6? What are the illegible markers in figure 6?

Lines 230-232 can not be understand.

Are the regions R_L and R_H also function of integration time as is I_T?

Lines 236-248 remain unclear. What is "image quality" (line 236)? Applies equation 4 only to the pixels within the range? "length", "height" (line 248)?

Lines 255-261 remain unclear. Make an equation for line 255-256! What are you bubble sorting? Make an equation for I_h (line 258) and I_l (line 261)!

Equation 10: What is M_eanl? Is it I_mean?

Chapter 2.3: Why not just put in all integration time images into MIF&DE?

Figure 9: Is it "seed points" or "seed point"?

Not really clear what you mean with "changes from (15,35) to (15,40)" in line 432.

Reviewer 2 Report

The authors have successfully answered all my concerns.

Author Response

Thanks for your viewing and comments.